# Therapeutic Targets and Emerging Treatments in Advanced Chondrosarcoma

**DOI:** 10.3390/ijms23031096

**Published:** 2022-01-20

**Authors:** Shinji Miwa, Norio Yamamoto, Katsuhiro Hayashi, Akihiko Takeuchi, Kentaro Igarashi, Hiroyuki Tsuchiya

**Affiliations:** Department of Orthopedic Surgery, Graduate School of Medical Science, Kanazawa University, Kanazawa 920-8640, Japan; norinori@med.kanazawa-u.ac.jp (N.Y.); hysk@med.kanazawa-u.ac.jp (K.H.); a_take@med.kanazawa-u.ac.jp (A.T.); kenken99004@med.kanazawa-u.ac.jp (K.I.); tsuchi@med.kanazawa-u.ac.jp (H.T.)

**Keywords:** chondrosarcoma, chemotherapy, immunotherapy, therapeutic target

## Abstract

**Simple Summary:**

Chondrosarcomas develop chemoresistance to standard anticancer drugs, making it difficult to control unresectable or metastatic chondrosarcomas. To improve the clinical outcomes of chondrosarcoma, new treatment approaches, such as molecule-targeting agents and immunotherapy, are needed. Recent research has revealed promising biomarkers and therapeutic targets for chondrosarcoma. In addition, several molecule-targeting agents have shown favorable antitumor activities in several clinical studies in patients with advanced sarcomas, including chondrosarcoma. This review summarizes recent basic studies on biomarkers and therapeutic targets and recent clinical studies on treating chondrosarcomas.

**Abstract:**

Due to resistance to standard anticancer agents, it is difficult to control the disease progression in patients with metastatic or unresectable chondrosarcoma. Novel therapeutic approaches, such as molecule-targeting drugs and immunotherapy, are required to improve clinical outcomes in patients with advanced chondrosarcoma. Recent studies have suggested several promising biomarkers and therapeutic targets for chondrosarcoma, including *IDH1/2* and *COL2A1*. Several molecule-targeting agents and immunotherapies have shown favorable antitumor activity in clinical studies in patients with advanced chondrosarcomas. This review summarizes recent basic studies on biomarkers and molecular targets and recent clinical studies on the treatment of chondrosarcomas.

## 1. Introduction

Chondrosarcoma, the second most common primary bone sarcoma, characterized by the production of the cartilaginous matrix, accounts for approximately 30% of malignant bone tumors [1]. According to the World Health Organization (WHO) classification, cartilaginous tumors are classified as benign, intermediate (locally aggressive or rarely metastasizing), and malignant tumors [2]. Conventional chondrosarcomas are classified into grades 1, 2, and 3 based on histological grades. Non-conventional chondrosarcomas include clear cell, periosteal, mesenchymal, and dedifferentiated chondrosarcomas. The 5-year survival rates of patients with low-grade, high-grade, and dedifferentiated chondrosarcomas are reported to be 83%, 53%, and 7–24%, respectively [3,4]. Most chondrosarcomas are resistant to chemotherapy and radiation therapy, unlike mesenchymal and dedifferentiated chondrosarcomas that are sensitive to these treatments [5,6,7,8]. Therefore, there are limited treatment options for patients with metastatic or unresectable chondrosarcomas.

Recent basic and clinical studies have investigated the efficacy and safety of new treatment modalities for advanced chondrosarcomas. This review discusses recent studies on gene mutations, biomarkers, anticancer agents, immunotherapy, and other promising treatments for chondrosarcomas.

## 2. Gene Mutations in Chondrosarcoma

Most chondrosarcomas are resistant to anticancer agents; therefore, identifying new therapeutic targets in chondrosarcoma is crucial. Gene mutations in chondrosarcomas can be helpful in investigating new therapeutic approaches and prognostic factors. Associations of several gene mutations with chondrosarcoma progression have been reported [9,10,11,12,13]. Chondrosarcomas frequently have gene mutations of isocitrate dehydrogenase 1/2 (*IDH1/2*), followed by collagen type II alpha 1 chain (*COL2A1*) and *TP53* (Table 1).

IDH, an NADP+-dependent enzyme, catalyzes the oxidative decarboxylation of isocitrate to α-ketoglutarate in the Krebs cycle. IDH1 exists in the cytoplasm, and IDH2 exists in the mitochondria. *IDH1/2* mutations are characteristic gene mutations that are detected in approximately 50% of chondrosarcomas and central and periosteal cartilaginous tumors [10]. These mutations result in the accumulation of 2-hydroxyglutarate (2HG), an oncometabolite [14,15]. Increased production of 2HG inhibits ten-eleven translocation (TET) enzymes, resulting in excessive methylation of DNA at CpG islands, and the regions of CpG island hypermethylation are enriched for genes related to stem cell maintenance, differentiation, and lineage specification [16]. Lu et al. reported that the expression of mutant *IDH2* caused DNA hypermethylation and impaired differentiation in murine 10T1/2 mesenchymal progenitor cells, the generation of undifferentiated sarcomas, and the hypermethylation and impairment in differentiation could be reversed by treatment with DNA hypomethylating agents [16]. In contrast, the influence of *IDH* mutations on tumorigenesis and progression in chondrosarcoma is controversial [17,18,19]. Li et al. reported that the IDH1 inhibitor AGI-5198 decreased oncometabolite D-2-hydroxyglutarate (D-2-HG), inhibited proliferation and migration, and induced apoptosis in chondrosarcoma cells [18]. In contrast, Suijker et al. reported that AGI-5198 dose-dependently decreased D-2-HG levels, although the study showed no significant influence on proliferation and migration [19].

Mutations of the major cartilage collagen gene *COL2A1*, with deletions, insertions, and rearrangements, were identified in 37% of chondrosarcomas [11]. The patterns of mutations suggest impairments in normal collagen biosynthesis in chondrosarcoma [11]. These mutations cause impairment of fundamental extracellular matrix (ECM) deposition and signaling, which may contribute to tumorigenesis by inhibiting normal cartilage differentiation [11]. Furthermore, the study showed mutations in *TP53* (20%), the RB1 pathway (33%), and Hedgehog signaling (18%) in chondrosarcoma [11].

*TP53* mutations are among the most frequently observed mutations in human cancers [20,21]. *TP53* mutations are observed in 20–50% of conventional chondrosarcomas and dedifferentiated chondrosarcomas [5,11,22,23,24]. Oshiro et al. reported a strong correlation between *TP53* mutations and metastatic disease or histological grade of chondrosarcoma [22]. Loss of function in *p53* may play an important role in the progression of chondrosarcoma and transformation to highly malignant chondrosarcomas [25,26,27].

Deletions in tuberous sclerosis 1 (*TSC1*) and phosphatase and tensin homolog (*PTEN*) genes and nonsense and missense mutations in the protein patched homolog 1 (*PTCH1*) have been observed in central chondrosarcomas [11,28]. Mutations in exostosin glycosyltransferase 1/2 (*EXT1/2*) genes, which participate in the differentiation of chondrocytes, have been observed in peripheral chondrosarcomas [11,26]. In a study using gene expression profiles obtained from the Gene Expression Omnibus database, differentially expressed genes between enchondromas and chondrosarcomas were investigated [12]. In this study, upregulated genes were related to epithelial–mesenchymal transition and the VEGF signaling pathway. The expression levels of four genes (*MFAP2*, *GOLM1*, *STMN1*, and *HN1*) increased continuously from control, enchondroma, to chondrosarcoma, and the expression of two genes (*CAB39L* and *GAB2*) decreased.

Although low levels of the tumor mutational burden (TMB) have been reported in chondrosarcoma [28], TMB is associated with histological grades. Grades 2 and 3 and dedifferentiated chondrosarcomas have levels of somatic mutations more than two times higher than grade 1 chondrosarcoma [11].

**Table 1 ijms-23-01096-t001:** Gene mutations in cartilaginous tumors.

Gene	Tumor Type	N	Frequency	Reference
*IDH1/2*	Cartilaginous tumors	220	50%	[10]
*IDH1/2*	Chondrosarcoma	488	*IDH1* 39%, *IDH2* 12%	[29]
*COL2A1*	Chondrosarcoma	49	37%	[11]
*TP53*	Chondrosarcoma	49	20%	[11]
*TSC1*	Chondrosarcoma	49	2%	[11]
*PTEN*	Chondrosarcoma	49	2%	[11]
*PTCH1*	Chondrosarcoma	49	8%	[11]
*EXT1/2*	Multiple osteochondroma	895	*EXT1* 65%, *EXT2* 35%	[30]

## 3. Biomarkers in Chondrosarcoma

The identification of biomarkers is beneficial in the management of malignancies because of various applications, including screening, differential diagnosis, the prediction of prognosis, and the monitoring of tumor progression. Several candidates of biomarkers for chondrosarcoma have been reported [31,32]. Mutations in *IDH1/2* have been considered biomarkers for acute myeloid leukemia and glioma [33,34]. In chondrosarcomas, Ollier disease, and Maffucci syndrome, a high incidence of *IDH1* and *IDH2* mutations has been reported [35]. In a meta-analysis of 488 patients with chondrosarcoma, *IDH1* and *IDH2* mutations were detected in 39% and 12% of patients, respectively [29]. In the study, *IDH1/2* mutations were correlated with age, origin, histological grade, tumor diameter, relapse, and mortality, and multivariate analysis revealed a significant association between *IDH1/2* mutations and poor overall survival. Nakagawa et al. investigated the correlation between *IDH* mutations and clinical outcomes in chondrosarcoma [36]. In the study, 15 (39%) of 38 patients had *IDH1* mutations, and 5 (13%) of 38 patients showed *IDH2* mutations. The study showed that *IDH* mutation was associated with worse overall survival, and *IDH* mutation was a significantly poor prognostic factor in univariate and multivariate analyses.

Giordano et al. investigated the expression of Eph type-A receptor (EphA2), a key oncoprotein implicated in angiogenesis, self-renewal, and metastasis, in bone sarcoma [37]. In the study, tumor tissues of osteosarcoma, Ewing sarcoma, and chondrosarcoma showed higher expression of EphA2 compared to normal tissues. Furthermore, the EphA2 inhibitor showed significant antitumor effects in patient-derived bone sarcoma cells. These data suggest that EphA2 is a potential therapeutic target in bone sarcoma, including chondrosarcoma.

Small ubiquitin-like modifier (SUMO), which is covalently attached to target proteins as a post-translational modification to alter the stability, localization, and function of the protein, can be a potential biomarker and therapeutic target in patients with chondrosarcoma. Kroonen et al. investigated the correlation between the expression of SUMO and clinical outcomes in patients with chondrosarcoma [38]. They reported that higher expressions of SUMO1 and SUMO2/3 were associated with an increased histological grade, and that a high expression of SUMO2/3 correlated with poor overall survival (OS). Furthermore, the SUMO E1 inhibitor ML792 reduced the cell proliferation and viability of chondrosarcoma cell lines in vitro. These results suggest that SUMO may be a potential therapeutic target in chondrosarcoma.

Takeuchi et al. investigated the expression of the receptor for endogenous secretory advanced glycation endproducts (esRAGE) and its ligand, high-mobility group box-1 (HMGB1), and their association with the histological grade in cartilaginous tumors [39]. In this study, the expression of esRAGE and HMGB1 was associated with the histological grade. Furthermore, the expression of esRAGE was associated with tumor recurrence, lung metastasis, and poor survival in patients with grade 1 chondrosarcoma. The results of the study suggest that esRAGE can be used as a biomarker to predict the histological grade and prognosis in patients with chondrosarcomas.

High expressions of the aurora kinases, which belong to the family of serine and threonine kinases, have been reported in several malignant tumors [40]. Liang et al. reported high expressions of aurora kinase A and B in high-grade chondrosarcoma compared to low-grade chondrosarcoma, and reported that the expression of aurora kinase correlated with worse survival in patients with chondrosarcomas [41].

Hypoxia-inducible factor (HIF) is an important transcription factor that contributes to cellular responses to hypoxic conditions, tumor survival, proliferation, and progression [42]. Chen et al. reported a high expression of HIF-2α in chondrosarcoma tissues compared to normal tissues [43]. In this study, the level of Beclin 1, a key mediator of autophagy, was significantly more decreased in chondrosarcoma tissues compared to normal tissues. HIF-2α and Beclin 1 had a significant inverse relationship with the prognosis in patients with chondrosarcoma. In another study, a high expression of HIF-1α was observed in patients with chondrosarcomas, and the increased expressions of HIF-1α were correlated with higher histological grades and clinical outcomes [44]. These findings suggest that HIF may be a prognostic predictor in patients with chondrosarcoma.

Rozeman LB et al. investigated the association between the expression of several molecules, including cyclin D1, p53, and plasminogen activator inhibitor 1 (PAI-1), with the patient survival [32]. In this study, the expression of PAI-1 correlated with better survival in patients with dedifferentiated peripheral chondrosarcomas.

The Hedgehog signaling pathway regulates cell proliferation during embryogenesis. Tiet et al. reported high expressions of Hedgehog target genes *PTCH1* and *GLI1* in chondrosarcomas, and that Hedgehog protein increased the cell proliferation of chondrosarcoma, and inhibitors of Hedgehog signaling decreased the proliferation [31].

Parafioriti et al. investigated the associations between miRNA and miRNA-regulated pathways with tumorigenesis in chondrosarcoma grades 1–3 [45]. While all grades showed similar expression profiles of miRNA, including miR-140-3p, significantly different expression profiles of miRNA were observed between grade 1 and grade 2/3 chondrosarcomas. The study suggests the contribution of miRNAs and their target pathway to the progression of chondrosarcoma.

In a study using multi-omics molecular profiles of chondrosarcoma, the acquisition of a proliferative state, the silencing of the 14q32 imprinted locus, and DNA methylation of IDH mutations were important to predict the histological malignancy and the clinical outcome [46]. Furthermore, a multi-omics classification, established by combining these molecular characteristics, was highly associated with OS in patients with chondrosarcoma.

Shi et al. investigated the association of DNA methylation and the transcription of immune-related genes with changes in the tumor microenvironment and the prognosis in patients with osteosarcoma [47]. In the study, immune-related DNA methylation patterns (IMPs), clinical outcomes, and tumor microenvironment characteristics in the patients were investigated, and an IMP-associated scoring model was constructed and evaluated in an independent patient cohort. The model may enable the prediction of prognosis and potential rationale for targeted therapy and immunotherapy in osteosarcoma. Studies on IMP-related scoring models for chondrosarcoma are sought to predict patient survival and therapeutic responses.

## 4. Anticancer Agents

Chemotherapy is commonly ineffective in chondrosarcoma, and there is no standard systemic treatment for conventional chondrosarcoma [48]. Reasons for the chemoresistance include slow proliferation, multidrug-resistance 1 gene P-glycoprotein expression, high expressions of Bcl-2 family proteins, resulting in the activation of anti-apoptotic and pro-survival pathways, and poor vascularity, resulting in the poor delivery of anticancer agents [49,50]. Furthermore, the rarity of the disease makes it difficult to conduct randomized clinical studies of anticancer agents. However, clinical studies are investigating the efficacy and safety of new anticancer agents against advanced chondrosarcomas (Table 2).

In a phase I study of ivosidenib (NCT02073994), a selective inhibitor of mutant IDH1, 21 patients with mutant *IDH1* advanced chondrosarcoma were treated with ivosidenib (100 mg twice daily to 1200 mg once daily). There were 11 (65%) patients with stable disease (SD), 6 (35%) patients with progressive disease (PD), and no patients with partial response (PR) or complete response (CR). The median progression-free survival (PFS) was 5.6 months [51]. Although 12 patients had adverse events of a grade of ≥3, only one event of hypophosphatemia was classified as a treatment-related adverse event by the investigator. Based on this study, ivosidenib is a promising treatment option for patients with mutant *IDH1* chondrosarcoma. Further clinical studies are required to assess the safety and efficacy of ivosidenib in many study patients.

Regorafenib is a multikinase inhibitor that targets VEGFR, PDGFR, PDGFR, c-kit, RET, and Raf. In a phase 2 study, the safety and efficacy of regorafenib were investigated in patients with advanced chondrosarcoma (NCT02389244) [52]. In the study, 13 of 24 (54%) patients treated with regorafenib were progression-free at 12 weeks, whereas 5 of 16 (31%) placebo patients were progression-free at 12 weeks. The medians PFS of the placebo group and regorafenib groups were 8 months and 20 months, respectively. Responses at 12 weeks were PR in 2 (8%) patients, SD in 11 (46%) patients, and PD in 10 (42%) patients. Among the study patients, severe treatment-related adverse events (grade ≥3), including hypertension (12%), diarrhea (8%), thrombocytopenia (8%), and asthenia (8%), were observed.

Pazopanib is a multitargeted tyrosine kinase inhibitor that targets VEGF, platelet-derived growth factor, and KIT. In a phase 2 study of pazopanib, the antitumor activity and safety were assessed in 47 patients with unresectable or metastatic chondrosarcomas (NCT01330966) [53]. In this study, 47 patients with chondrosarcoma underwent pazopanib treatment (800 mg, daily). The disease control rate at 16 weeks was 43%. Among the 42 evaluable patients, no patient had CR, 1 patient (2%) had PR, 30 patients (64%) had SD, and 11 (26%) patients had PD at 16 weeks. The mean OS and PFS periods were 18 months and 8 months, respectively. Adverse events of grade 3 or higher, including hypertension (26%), elevated alanine aminotransferase (9%), neutropenia (4%), and pulmonary emboli (4%), were observed in this study.

Extraskeletal myxoid chondrosarcoma (ESMS) is a rare type of chondrosarcoma that commonly exhibits resistance to chemotherapy. In a phase 2, multicenter study of pazopanib in patients with metastatic or unresectable ESMS, the antitumor activity and safety of pazopanib were investigated (NCT02066285) [54]. In this study, 26 patients with ESMS received oral pazopanib (800 mg/day). Of the 22 evaluable patients, four (18%) patients had objective responses. The most frequent grade 3 adverse events were hypertension (35%) and increased alanine aminotransferase (23%) and aspartate aminotransferase (19%) levels. The results of the study indicate that pazopanib has meaningful activity against advanced ESMC and could be considered one of the treatment options in patients with ESMC.

Nilotinib is a selective tyrosine kinase inhibitor that targets BCR-ABL, c-KIT, PDGFR, and EGFR. Alemany et al. reported a phase 1 study of nilotinib plus doxorubicin in 13 patients with sarcoma, including seven chondrosarcomas, four liposarcomas, and two leiomyosarcomas (NCT02587169) [55]. The study patients were treated with nilotinib (400 mg, twice a day, days 1–6) and doxorubicin (60–75 mg/m^2^, day 5) every 3 weeks for 4 cycles. In the study, grade 3–4 treatment-related adverse effects included neutropenia (54%), febrile neutropenia (15%), and asthenia (8%). One patient (8%) (1 liposarcoma) showed PR, 9 (60%) patients had SD (5 chondrosarcomas, 2 liposarcomas, and 1 leiomyosarcoma), and 3 (23%) patients showed PD (two chondrosarcomas and one leiomyosarcoma).

Although chondrosarcomas commonly develop chemoresistance, several clinical studies have shown positive drug activities for chondrosarcoma. These clinical studies suggest that some molecule-targeting agents may have higher antitumor activity than conventional antitumor agents in chondrosarcoma. Further clinical studies of new anticancer agents are required to assess the efficacy and safety of the treatment in patients with chondrosarcoma.

Nilotinib is a selective tyrosine kinase inhibitor that targets BCR-ABL, c-KIT, PDGFR, and EGFR. Alemany et al. reported a phase 1 study of nilotinib plus doxorubicin in 13 patients with sarcoma, including seven chondrosarcomas, four liposarcomas, and two leiomyosarcomas (NCT02587169). The study patients were treated with nilotinib (400 mg, twice a day, days 1–6) and doxorubicin (60–75 mg/m^2^, day 5) every 3 weeks for 4 cycles. In the study, grade 3–4 treatment-related adverse effects included neutropenia (54%), febrile neutropenia (15%), and asthenia (8%). One patient (8%) (1 liposarcoma) showed PR, 9 (60%) patients had SD (5 chondrosarcomas, 2 liposarcomas, and 1 leiomyosarcoma), and 3 (23%) patients showed PD (two chondrosarcomas and one leiomyosarcoma).

## 5. Immunotherapy

Several immunohistochemical studies have shown immune conditions in chondrosarcoma [56,57,58,59]. Chondrosarcoma tissues showed an increased expression of PD-1 compared to normal bone and osteochondromas [56]. Kostine reported that an increased expression of PD-L1 was observed in 41% of dedifferentiated chondrosarcomas, and it was significantly correlated with TILs and HLA class I expression [57]. Yang X et al. reported that the positivity for the expressions of PD-L1 and PD-L2 in chondrosarcoma were 68% and 42%, respectively, and that the expression of PD-L1 was associated with young age, a large tumor size, histological grade, and tumor recurrence [58]. One study suggested the association of tumor-associated macrophages (TAM) with the microenvironment in chondrosarcoma [60]. Richert et al. reported that a high number of CD68+ macrophages were associated with metastases at diagnosis and poor outcomes [60]. In this study, increased expressions of lymphocyte activation gene-3 (LAG3), T cell immunoglobulin mucin (TIM3), B7 superfamily member-H3 (B7H3), signal regulatory protein alpha (SIRPA), and colony-stimulating factor 1 receptor (CSF1R) were observed in chondrosarcoma [60]. These proteins play important roles in the differentiation and survival of macrophages, the prevention of phagocytosis, the inhibition of T cell activity, and the influence of macrophage activity, migration, invasion, and angiogenesis in several malignancies [61,62,63,64,65]. Simard et al. reported that the infiltration of CD163+ macrophages was associated with the histological grade of chondrosarcoma. A high concentration of CD8+ T cells was associated with the inhibition of chondrosarcoma progression [66]. Kostine et al. reported that 47% of dedifferentiated chondrosarcomas showed the expression of PD-L1 and that it was correlated with a high number of tumor-infiltrating lymphocytes (TILs) and the expression of HLA class I [57].

In a phase 2 study (SARC028 trial, NCT02301039), 86 patients with advanced sarcomas underwent treatment with pembrolizumab (200 mg, intravenously, every 3 weeks, Table 3) [67]. The most frequent adverse events of grade 3 or higher, including anemia (14%), decreased lymphocyte count (12%), prolonged activated partial thromboplastin time (10%), and decreased lymphocyte count (7%) were observed in patients with bone sarcoma. Five patients with chondrosarcoma were included in the study, one patient (20%) had PR, one patient (20%) had SD, and three patients (60%) had PD.

In a retrospective study, the clinical outcomes of nivolumab were investigated in 28 patients with metastatic/unresectable sarcomas. In this study, 28 patients with bone and sarcomas were treated with nivolumab (3 mg/kg, intravenously, every 2 weeks) with or without pazopanib [68]. Four patients treated with nivolumab and pazopanib had grade 3–4 adverse events, including liver function test abnormalities, diarrhea, and pneumonitis, whereas none of the patients treated with only nivolumab had grade 3–4 adverse events. Among the 24 evaluable patients, 3 patients (13%), including dedifferentiated chondrosarcoma, epithelioid sarcoma, and osteosarcoma, had a partial response, 9 patients (38%) had stable disease, and 12 patients (50%) had disease progression [68].

Interleukin-8 (IL-8) is a chemokine that promotes epithelial-mesenchymal transition, immune escape, the recruitment of MDSCs, and tumor progression [69,70,71,72,73,74]. Bilusic et al. conducted a phase 1 study of BMS-986253, a human monoclonal antibody that inhibits IL-8, in patients with metastatic or unresectable solid tumors (NCT02536469) [75]. Fifteen patients, including one patient with chondrosarcoma, were treated with BMS-986253 (4, 8, 16, or 32 mg/kg, i.v., every 2 weeks). Five patients had grade 1 or 2 treatment-related adverse events, and none of them had severe treatment-related adverse events. Eleven (73%) patients had SD and 4 (27%) patients, including patients with chondrosarcoma, had PD. Although no objective response was observed in this study, a high rate of disease control was observed. Further studies including patients with chondrosarcoma are required to assess the efficacy in patients with chondrosarcoma.

**Table 3 ijms-23-01096-t003:** Clinical studies of immunotherapy for sarcomas.

Agent	Number of Patients	Phase	Diagnosis	Responses	Severe Adverse Events	References
Pembrolizumab (200 mg, iv, every 3 weeks)	86	2	Advanced sarcoma	Response rates: UPS (40%), liposarcoma (20%), SS (10%), LMS (0%), osteosarcoma (5%), chondrosarcoma (20%), and ES (0%)	Anemia (14%), decreased lymphocyte count (12%), prolonged activated partial thromboplastin time (10%), and decreased lymphocyte count (7%)	[67]
BMS-986253 (4, 8, 16, or 32 mg/kg, i.v., every 2 weeks)	15	1	Metastatic or unresectable solid tumors	SD 73% and PD 27%	No patient had severe treatment-related adverse event	[75]
Nivolumab (3 mg/kg, intravenous, every 2 weeks) with or without pazopanib	28	Retrospective study	Metastatic of unresectable sarcomas	PR 13%, SD 38%, PD 50%	Liver function abnormalities (11%), diarrhea (4%), and pneumonitis (4%)	[68]

UPS, undifferentiated pleomorphic sarcoma; SS, synovial sarcoma; LMS, leiomyosarcoma; ES, Ewing sarcoma; SD, stable disease; PD, progressive disease: PR, partial response.

## 6. Promising Candidates for Therapeutic Targets

Although no marked development in the treatment of chondrosarcoma has been observed, recent basic studies on the molecular mechanisms underlying tumor progression in chondrosarcomas have suggested promising therapeutic targets.

The ligand-activated transcription factor peroxisome proliferator-activated receptor gamma (PPARγ) has been reported as a candidate for a therapeutic target in malignant tumors, including chondrosarcomas [76,77]. Nishida et al. reported that PPARγ dose-dependently inhibited cell proliferation and induced apoptosis in chondrosarcoma cells [76]. Higuchi et al. reported that zaltoprofen, a nonsteroidal anti-inflammatory drug, could activate PPARγ in chondrosarcoma cells [77]. In the study, zaltoprofen inhibited cell proliferation, viability, migration, and invasion via PPARγ activation and matrix metalloproteinase-2 activation.

Ouyang et al. reported that a high expression of CDK4 was observed in chondrosarcoma samples and cell lines, and the expression of CDK4 was associated with metastasis and poor prognosis [78]. In this study, the proliferation of chondrosarcoma cells was decreased by inhibiting CDK4 by siRNA. Palbociclib, an inhibitor of CDK4, induced cell cycle arrest in the G1 phase and inhibited the proliferation, migration, and invasion of chondrosarcoma cells through the CDK4/Rb signaling pathway.

NEDD8 is a ubiquitin-like protein that covalently conjugates with cellular proteins. It has been reported that MLN4924, a selective inhibitor of NEDD8-activating enzyme, is a promising candidate for novel anticancer treatment [79]. Wu et al. investigated the antitumor effects of MLN4924 on chondrosarcoma cells [80]. In this study, MLN4924 inhibited the cell proliferation, cytotoxicity, and induction of apoptosis in chondrosarcoma cell lines, and inhibited tumor growth in a xenograft mouse model of chondrosarcoma.

Resveratrol, a dietary phytochemical found in various plant species, has shown antitumor effects on several types of malignant tumors [81]. Jin et al. investigated the effect of resveratrol on tumor growth and sirtuin 1 activity in chondrosarcoma cells [82]. They reported that resveratrol showed antitumor effects, including the inhibition of cell viability and proliferation, the induction of apoptosis, and the suppression of phosphorylation within the STAT3 signaling pathway, in chondrosarcoma cells.

The mammalian target of rapamycin (mTOR) plays an important role in cell proliferation, apoptosis, and autophagy [83]. The mTOR inhibitor rapamycin reduced oxidative and glycolytic metabolism and dose-dependently decreased cell viability in chondrosarcoma cell lines [84,85]. In another study, the mTOR inhibitor everolimus suppressed the expressions of Glut1 and HIF1α, cell proliferation, and prevented tumor progression in a rat orthotopic chondrosarcoma model [84].

## 7. Conclusions

Recent studies have suggested several promising biomarkers and therapeutic targets for chondrosarcoma, including IDH1/2, COL2A1, and PD-L1. In addition, several molecule-targeting agents and immunotherapy have shown favorable antitumor activities in clinical studies of patients with advanced chondrosarcoma.

Further studies on the mechanisms of tumor proliferation, invasion, migration, and microenvironment are required to identify new therapeutic targets in chondrosarcoma. Basic studies and clinical trials of new anticancer agents and immunotherapy may contribute to the development of chondrosarcoma treatments.

## Figures and Tables

**Table 2 ijms-23-01096-t002:** Clinical studies and target molecules for chondrosarcomas.

Treatment	Target Molecule	N	Phase	Tumor Type	Clinical Significance	Grade 3–4 Toxicities	References
Ivosidenib (100 mg twice daily to 1200 mg once daily)	IDH1	21	1	Advanced chondrosarcoma	SD 65%, PD 35%; PFS: 5.6 months	Edema (5%), pain in extremity (5%), anemia (5%), and increased alkaline phosphatase (5%)	[51]
Regorafenib (daily, 160 mg)	VEGFR, PDGFR, PDGFR, c-kit, RET, Raf	24	2	Advanced chondrosarcoma	PFS: 20 (regorafenib) and 8 (placebo) months)	Hypertension (12%), diarrhea (8%), thrombocytopenia (8%), and asthenia (8%)	[52]
Pazopanib (daily, 800 mg)	VEGF-1, 2, 3 PDGFR, c-kit	47	2	Unresectable or metastatic chondrosarcomas	PR 2%, SD 64%, PD 26%; PFS: 8 months; OS: 18 months	Hypertension (26%) and elevated alanine aminotransferase (9%), neutropenia (4%), and pulmonary emboli (4%)	[53]
Pazopanib (daily, 800 mg)	VEGF-1, 2, 3 PDGFR, c-kit	26	2	Metastatic or unresectable ESMS	Objective response: 18%	Hypertension (35%), increased alanine aminotransferase 23%), and increased aspartate aminotransferase (19%)	[54]
Nilotinib (day 1–6, 400 mg/12 h) and doxorubicin (60–75 mg/m^2^, day 5), every 3 weeks	BCR-ABL, c-KIT, PDGFR, EGFR	13	1	Retroperitoneal liposarcoma, leiomyosarcoma, and advanced chondrosarcoma	PR 8%, SD 69%, PD 23%	Neutropenia (54%), febrile neutropenia (15%), and asthenia (8%)	[55]

IDH, isocitrate dehydrogenase; VEGFR, vascular endothelial growth factor receptor; PDGFR, platelet-derived growth factor receptor; SD, stable disease; PD, progressive disease; PR, partial response; PFS, progression-free survival; OS, overall survival.

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
