# Peer review of "Therapeutic Targets and Emerging Treatments in Advanced Chondrosarcoma"

_ijms, 2022, doi:10.3390/ijms23031096_

Round 1

Reviewer 1 Report

This is a well conducted review presented in a well-written manuscript. 

I have only one comment: If possible, I would prefer more tables in order to make it easier for the readers to get a better and faster overview regarding all the data that are included in this review.   

Author Response

We sincerely appreciate you for your considering our manuscript entitled “Therapeutic targets and emerging treatments in advanced chondrosarcoma” for publication in International Journal of Molecular Sciences. We are very grateful for your prompt attention and thorough review.

Tables including the data were added to the manuscript.

Reviewer 2 Report

The Authors provide an accurate review on the biomarkers and therapeutic targets of chondrosarcoma. The review is well organized and carefully written. However, to further improve their description the authors should provided some comments about novel additional biormakers and molecular classifications. This include:

  • The molecular classification made by Nicolle and colleagues (doi: 10.1038/s41467-019-12525-7) and the one proposed by Deyao et al., (doi: 10.1002/1878-0261.13160).
  • The description of novel molecular biomarkers, including Epha2 (doi: 10.3390/cells10112893), the SUMOylation process (doi: 10.3390/cancers13153823), and noncoding RNAs like miRNAs (doi: 10.1038/s41420-020-0282-3).
  • Adding a table summaryzing the evidence supporting evidence of each gene alteration with indication of the sample size used in each study will be valuable to prioritize  the information available on each gene.

Author Response

We sincerely appreciate you for your considering our manuscript entitled “Therapeutic targets and emerging treatments in advanced chondrosarcoma” for publication in International Journal of Molecular Sciences. We are very grateful for your prompt attention and thorough review.

The following sentences and table were added to the manuscript.

                 Giordano et al. investigated expression of Eph type-A receptor (EphA2), a key oncoprotein implicated in angiogenesis, self-renewal, and metastasis, in bone sarcoma 1. In the study, tumor tissues of osteosarcoma, Ewing sarcoma, and chondrosarcoma showed higher expression of EphA2 compared to normal tissues. Furthermore, EphA2 inhibitor showed significant antitumor effect in patient-derived bone sarcoma cells. These data suggests that EphA2 is considered as potential therapeutic target in bone sarcoma including chondrosarcoma.

                 Small ubiquitin-like modifier (SUMO), which is covalently attached to target proteins as a post-translational modification to alter the stability, localization, and function of the protein, can be a potential biomarker and therapeutic target in patients with chondrosarcoma. Kroonen et al. investigated correlation between expression of SUMO and clinical outcomes in patients with chondrosarcoma 2. They reported that higher expression of SUMO1 and SUMO2/3 was associated with increased histological grade, and that high expression of SUMO2/3 correlated with poor overall survival (OS). Furthermore, SUMO E1 inhibitor ML792 reduced cell proliferation and viability of chondrosarcoma cell lines in vitro. These results suggest that SUMO may be a potential therapeutic target in chondrosarcoma.

                 Parafioriti et al. investigated the associations between miRNA and miRNA-regulated pathways with tumorigenesis in grade 1–3 chondrosarcoma 3. While all grades showed similar expression profile of miRNA including miR-140-3p, significantly different expression profile of miRNA was observed between grade 1 and grade 2/3 chondrosarcomas. The study suggests the contribution of miRNA and their target pathway to the progression of chondrosarcoma.

                 In a study using multi-omics molecular profiles of chondrosarcoma, acquisition of a proliferative state, silencing of the 14q32 imprinted locus, and DNA methylation of IDH mutations, were important to predict the histological malignancy and the clinical outcome 4. Furthermore, a multi-omics classification established by combining these molecular characteristics, was highly associated with OS in patients with chondrosarcoma.

                 Shi et al. investigated the association of DNA methylation and transcription of immune-related genes with changes in the tumor microenvironment and prognosis in patients with osteosarcoma 5. In the study, immune-related DNA methylation patterns (IMPs), clinical outcomes, and tumor microenvironment characteristics in the patients were investigated, and an IMP-associated scoring model was constructed and evaluated in an independent patient cohort. The model may enable prediction of prognosis and potential rationale for targeted therapy and immunotherapy in osteosarcoma. The study on IMP-related scoring model for chondrosarcoma is demanded to predict patient survival and therapeutic responses.

References

Giordano, G.;  Merlini, A.;  Ferrero, G.;  Mesiano, G.;  Fiorino, E.;  Brusco, S.;  Centomo, M. L.;  Leuci, V.;  D'Ambrosio, L.;  Aglietta, M.;  Sangiolo, D.;  Grignani, G.; Pignochino, Y., EphA2 Expression in Bone Sarcomas: Bioinformatic Analyses and Preclinical Characterization in Patient-Derived Models of Osteosarcoma, Ewing's Sarcoma and Chondrosarcoma. Cells 2021, 10 (11).

Kroonen, J. S.;  Kruisselbrink, A. B.;  Briaire-de Bruijn, I. H.;  Olaofe, O. O.;  Bovee, J.; Vertegaal, A. C. O., SUMOylation Is Associated with Aggressive Behavior in Chondrosarcoma of Bone. Cancers (Basel) 2021, 13 (15).

Parafioriti, A.;  Cifola, I.;  Gissi, C.;  Pinatel, E.;  Vilardo, L.;  Armiraglio, E.;  Di Bernardo, A.;  Daolio, P. A.;  Felsani, A.;  D'Agnano, I.; Berardi, A. C., Expression profiling of microRNAs and isomiRs in conventional central chondrosarcoma. Cell Death Discov 2020, 6, 46.

Nicolle, R.;  Ayadi, M.;  Gomez-Brouchet, A.;  Armenoult, L.;  Banneau, G.;  Elarouci, N.;  Tallegas, M.;  Decouvelaere, A. V.;  Aubert, S.;  Redini, F.;  Marie, B.;  Labit-Bouvier, C.;  Reina, N.;  Karanian, M.;  le Nail, L. R.;  Anract, P.;  Gouin, F.;  Larousserie, F.;  de Reynies, A.; de Pinieux, G., Integrated molecular characterization of chondrosarcoma reveals critical determinants of disease progression. Nat Commun 2019, 10 (1), 4622.

Shi, D.;  Mu, S.;  Pu, F.;  Liu, J.;  Zhong, B.;  Hu, B.;  Ni, N.;  Wang, H.;  Luu, H. H.;  Haydon, R. C.;  Shen, L.;  Zhang, Z.;  He, T. C.; Shao, Z., Integrative analysis of immune-related multi-omics profiles identifies distinct prognosis and tumor microenvironment patterns in osteosarcoma. Mol Oncol 2021.